# The Outcomes of an Interprofessional Simulation Program for New Graduate Nurses

**DOI:** 10.3390/ijerph192113839

**Published:** 2022-10-25

**Authors:** Shu-Ling Yeh, Chiu-Tzu Lin, Li-Hsiang Wang, Chun-Chih Lin, Chen-Te Ma, Chin-Yen Han

**Affiliations:** 1Department of Nursing, Chang Gung Memorial Hospital at Keelung, Keelung 20401, Taiwan; 2Department of Nursing, Chang Gung University of Science and Technology, Taoyuan 33303, Taiwan; 3Department of Nursing, Chang Gung Memorial Hospital at Linkou, Taoyuan 333423, Taiwan; 4Department of Nursing, New Taipei Municipal TuCheng Hospital, Chang Gung Medical Foundation, New Taipei City 23652, Taiwan

**Keywords:** interprofessional education, simulation training, new graduate nurses, core competency

## Abstract

This study explored the learning outcomes of an interprofessional simulation program for new graduate nurses during their training program. It was a single-group, pre-test and post-test research design. Ninety-three new graduate nurses participated in the study. The Nursing Competence Instrument and program satisfaction survey questionnaires were used to evaluate the learning outcomes of the program. Data were collected between 1 July 2019 and 30 June 2020 in a medical center in Taiwan. It was found that four nursing core competencies were significantly higher after the simulation, including advancing career talents (t = 10.12, *p* < 0.0001), integrating care abilities (t = 10.19, *p* < 0.001), dealing with tension (t = 6.87, *p* < 0.0001), and leading humanity concerns (t = 6.86, *p* < 0.001). The average satisfaction score for the interprofessional simulation training among nurses was 4.42 out of 5. In conclusion, interprofessional simulation education can help novice nurses improve their nursing core competencies. The results of this study provide an important indicator for hospitals and governments when making policy and training programs for new graduate nurses.

## 1. Introduction

Newly graduated nurses often need additional training to improve their ability to assess the care needs of patients who have complex illnesses and psychosocial problems, and to engage in interprofessional collaboration. Interprofessional collaboration is an important core competence for nurses [1], which enhances nurses’ ability to develop teamwork in patient care and collaborative care with other healthcare professionals [2,3,4]. An important component of interprofessional education is simulation training, which helps develop professional skills and collaboration among nurses and other care givers [5,6]. Interprofessional simulation training also can help close the gap between academic learning and clinical nursing practice for new graduate nurses.

It was reported that new graduate nurses are concerned about their ability to care for patients in collaboration with other healthcare professionals [7,8,9]. The literature indicates that new graduate nurses experience difficulty meeting clinical expectations. This problem can be addressed through simulation-based education, which improves clinical competence and practice [10]. New graduate nurses encounter numerous challenges to their professional skills, responsibilities in patient care, and the need to interact well with other professionals [9,11,12]. The gap between medical knowledge and skills needed in clinical practice may challenge new graduate nurses when accommodating themselves to their new roles [11,12]. Interprofessional education provides new graduate nurses an opportunity to boost their hands-on capacity, as good practical skills enhance confidence in patient care.

The majority of the nursing curriculum and the training taught in classrooms contributes to the theory–practice gap and does not address issues in clinical practice [13,14]. As a result, new graduate nurses often have difficulty in applying their new knowledge to clinical settings. Developing the ability to collaborate with others in patients’ care is one of the goals of new-graduate-nurse training. This type of training helps individual healthcare workers develop the skills required to establish safe, timely, efficient, and effective medical care systems in collaboration with other healthcare providers [3,15].

The Ministry of Health and Welfare in Taiwan initiated the teaching reimbursement plan in 2009. The plan was intended to encourage hospitals to provide learning resources and training environments to aid the development of the core competencies of healthcare professionals. The reimbursement plan provided financial support for a two-year initiative for registered nurses licensed within the first two years [15]. The Taiwan Nursing Accreditation Council (TNAC) has proposed that nursing education nurture the development of students’ and nurses’ core competencies for the nursing profession. The nursing core competencies define what is needed for a nurse to care for patients competently, consisting of the composite outcomes of knowledge, skills, common practice, and attitudes [16]. The core competencies should be closely linked to clinical practice because they represent the essential skills of competent nurses, and high-value clinical care is the goal of nursing education [17]. The core competencies of nursing education in Taiwan include critical thinking and reasoning, general clinical skills, basic biomedical science, communication and teamwork capability, caring, ethics, accountability, and lifelong learning [18].

Simulation teaching plays a significant role in the development of awareness and professional skills in interprofessional care [7,19,20]. Simulation-based teaching provides new graduate nurses with the ability to learn through safe, simulated scenarios. It helps to improve familiarity with the clinical setting [21], improve interprofessional communication skills and work satisfaction [22], increase understanding of other healthcare professionals’ responsibilities, encourage caring interprofessional attitudes, promote critical thinking [20,22], develop the ability to adapt and counteract, advance the recognition of patient safety [23], and inspire learning motives and interests [24]. Hospital training programs for new graduate nurses provide an ideal environment for examining the impact of simulations on the development of professional skills and interprofessional collaboration. Hospitals in Taiwan have formal nursing training programs in the new graduate year that are supported by the Ministry of Health and Welfare. The simulations in these programs can be adjusted to help new graduate nurses develop interprofessional collaboration abilities. The aim of this study was to assess the outcomes of interprofessional simulation training program for new graduate nurses in Taiwan.

## 2. Materials and Methods

### 2.1. Design

This was a single-group, pre-test and post-test design conducted in a 3000-bed hospital in Taiwan for which approximately 250 new graduate nurses are recruited every year.

### 2.2. Sample

Purposive sampling was used to select participants who were >20 years of age, had recently graduated from nursing programs, and were attending new-graduate-nurses training. Exclusion criteria included having worked as a nurse for more than two years or having transferred from another hospital. Participants were recruited between 1 July 2019 and 30 June 2020.

### 2.3. Measurements

#### 2.3.1. Demographic Data

A personal information questionnaire was used to collect each participant’s characteristics, including gender, age, level of education, the department in which they worked, and years of nursing experience.

#### 2.3.2. Nursing Competence

Nursing competence was measured using the Nursing Competence Instrument (NCI). The NCI [25] is a self-administrated questionnaire containing 27 items across four subscales: integrating care abilities (ability based on knowledge synthesis and critical thinking), leading humanity concerns (providing caring, such as compassionate care and empathy), advancing career talents (ongoing professional development of skills), and dealing with tension (recognizing and coping with stressors). The NCI has 10 items for the integrating care abilities subscale, including critical thinking, communication, and teamwork; six items for the leading humanity concerns subscale, including accountability; seven items for the advancing career talents subscale; and four items for the dealing with tension subscale, including lifelong learning, stress, and coping. Participants rated each item on a 10-point Likert scale, from the highest score (10 points) to the lowest score (1 point), to indicate the extent of their agreement with each item. The content consistency in the NCI achieved a value of 0.96. The internal consistency reliability in this study was high, with a Cronbach’s alpha value of 0.88. The validity of the evaluation was assessed by three experts. The necessary adjustments were made before the survey was completed.

The development of the interprofessional simulation program was based on Taiwan’s TNAC standard of nursing core competency. The NCI tool was chosen for the present study because it was developed based on the TNAC’s core competencies for the nursing profession in Taiwan. The literature indicates a need for individual countries to develop assessment tools based on national guidelines [26]. Therefore, the NCI is a suitable tool for evaluating the core competencies as learning outcomes for new graduate nurses in Taiwan.

#### 2.3.3. Learning Satisfaction

The research team developed the Learning Satisfaction Questionnaire based on an integrated literature review to allow participants to self-assess their satisfaction with the simulation training. The 14 items in the questionnaire measured aspects of learning satisfaction related to the content and teaching strategy of the program, including practicality, suitability, and clinical applicability. Each item was scored on a 5-point Likert scale with the following values: highly agree (5 points), agree (4 points), neither agree nor disagree (3 points), disagree (2 points), and highly disagree (1 point). Participants were asked to read the questionnaire and circle the answer that best described their degree of agreement with each item. The questionnaire also included two open-ended questions that allowed participants to express their experiences of or suggestions for the program. The reason for using the Learning Satisfaction Questionnaire in the study was to let the participants express their satisfaction and comments related to the content and teaching strategy of the interprofessional simulation program.

#### 2.3.4. Simulation Program

The interprofessional simulation program follows a competency-based design and derives from Taiwan’s TNAC standard of core nursing competency. The aim of the interprofessional simulation program in this study is to enhance four nursing competencies—knowledge, skills, caring, and dealing with tension—that new graduate nurses should demonstrate while providing multidisciplinary care to patients presenting emergency conditions. The case scenario is of a patient with chest tube drainage and blood transfusion who suddenly presents with dyspnea and hypotension. The new graduate nurse participants need to provide an initial assessment, contact physicians and respiratory therapists, and go through the entire resuscitation process with the team to stabilize the patient. As family members are frequently primary caregivers in Taiwan, the participants must also take care of anxious family members during the simulation program. The interprofessional simulation program has five learning objectives: (1) the identification of clinical problems; (2) the management of clinical problems; (3) the application of communication skills and participation in collaborative care with a multidisciplinary team member; (4) the demonstration of compassionate care and communication skills with the patient and involved family member; and (5) the evaluation of the care results. Table 1 details how nursing competency is aligned with the learning objectives of the interprofessional simulation program.

Clinical cases are recommended for use in scenarios intended to train learners in common team capabilities [27,28]. The patient scenarios selected for this study were based on clinical cases that occurred in the participating hospital in the previous year. Handouts were provided that contained basic patient information and learning objectives [29]. The scenario curriculum included prior knowledge and learning goals associated with knowledge and skills [30]. Trained actors portrayed patients, family members, nurses, physicians, and respiratory therapists and represented the most common combinations of multidisciplinary teams that treat patients, respond to emergency situations, and deal with the emotions of patients and family members [31]. Each simulation had three facilitators on stage to conduct the simulation, communicate instructions, and handle common problems in interprofessional communication encountered during the discussion of clinical cases.

Five experts, including a senior nurse, a head nurse, a nurse educator, and two university instructors, completed the validity assessment of the simulation scenarios. The experts evaluated the content of simulation scenarios based on their applicability, necessity, clarity of expression, and content coverage. The scoring standards were highly inapplicable (1 point), inapplicable (2 points), neither applicable nor inapplicable (3 points), applicable (4 points), and highly applicable (5 points). The simulation scenarios were adjusted after a first assessment before being given to the three experts for a second assessment. A content validity index value >80% was taken as evidence that the experts deemed that the scenarios had suitable content validity. To assess instructor consistency and discuss the consistency of the scenario content, the research team held a consistency meeting for each training session prior to implementing the scenario. The discussion covered areas such as learning goals, teaching methods, questions, discussion, and debriefing.

#### 2.3.5. Procedure

This study was approved by the institutional review board of the participating hospital (IRB number 201702115B0). All participants were volunteers who were informed that their withdrawal from the research at any time would not affect their work status. Written informed consent was obtained from all participants in the study. Before entering the interprofessional simulation program, participants were required to complete the demographic questionnaire and the NCI. The interprofessional simulation training program was composed of a five-minute introduction, a 20 min simulation session, and a closing 15 min debriefing. The debriefing section aimed to guide all participants in retrospective learning and in reviewing the suitability, priority, and comprehensiveness of their actions before the next group of participants went on stage with the same scenario. Each workshop included three sessions dedicated to providing further opportunities for participants to practice and observe. Participants completed a post-test NCI at the end of the workshop. After completing the interprofessional simulation program, the participants also completed a learning satisfaction questionnaire. Using the NCI as a measurement tool helped the researchers understand and evaluate changes in the participants’ nursing competency related to critical thinking, communication, teamwork, accountability, empathy, and coping with stress during the simulation program. The participants could also observe their improvement before and after the simulation program through the NCI questionnaire. The Learning Satisfaction Questionnaire allowed the participants to express their experiences of and comments on the simulation program.

### 2.4. Statistical Analysis

All analyses were performed with the IBM SPSS Statistics for Windows Ver. 22 (IBM Corp., Armonk, NY, USA). Continuous variables were described using mean and standard deviations. Categorical variables were described using frequencies and percentages. Paired t-tests were used to analyze the NCI scores before and after the program. Pearson correlation was utilized to analyze the correlation among changes of NCI score. All tests were two-tailed, with *p* < 0.05 considered significant.

## 3. Results

### 3.1. Participants

Table 2 presents the demographic data of the nurse participants in the study. The study had three interprofessional simulation programs, with 96 new graduate nurses attending. Ninety-three nurses completed the questionnaires, for a response rate of 96.9%. The participants were primarily women (93.5%), aged 20–25 years, with an average age of 22.5 years (SD = 0.80). Almost all (95.7%) participants had a bachelor’s degree. Most of the participants worked in a general medical-surgical ward (60.2%) or in intensive care units (22.6%). Their experience in nursing ranged from 1–23 months, with an average of 7.0 months (SD = 5.42).

### 3.2. Change of Nursing Competence

This study used the NCI to evaluate the nursing core competency of participating new graduate nurses after completing their interprofessional simulation training. The results given in Table 3 indicate the significant differences between the pretest and post-test scores in all four areas: dealing with tension, integrating care abilities, leading humanity concerns, and advancing career goals.

In the comparison of pretest with post-test values based on work experience, we used nonparametric analysis for the nine participants whose work experience exceeded 12 months. The results (Table 4) revealed significant differences in dealing with tension, integrating care abilities, and advancing career talents between pretest and post-test scores; the amount of change in each ability and satisfaction displayed no difference. To further analyze the differences in core abilities, amount of change, and satisfaction of those, by the department in which they worked, we used nonparametric analysis due to the differences in the number of participants by department. The results revealed no difference among departments.

### 3.3. Learning Satisfaction and Qualitative Outcomes of the Interprofessional Simulation Program

The average learning satisfaction score for the interprofessional simulation training was 4.42 out of 5. The participants were most satisfied with the interprofessional simulation program to the extent that they perceived it as closely reflecting the clinical setting. The results of the qualitative feedback indicated that the participants’ learning outcomes addressed the four nursing competencies of care ability, advanced skills, caring, and dealing with tension, as summarized below. Participants expressed being able to identify insufficiencies in their prior knowledge and skills and reported enhanced preparation after completing the training. They stated that the interprofessional simulation training helped them transfer what they had learned in the scenarios to patient care and that the simulation scenarios were commonly reflected in their daily practice. The simulation scenarios increased their skills in dealing with patients who had emergency conditions. Participants also reported learning to use critical thinking to assess and evaluate the cause of a patient’s condition and to determine a solution, taking patient safety into consideration. In other qualitative feedback from the simulation training, the participants stated that they had become more familiar with other healthcare professionals and better understood the principles of communication and collaboration with other teams. The participants also felt that the training program improved their understanding of the practices that could be applied in patient care. Through the debriefing sessions, the participants reported receiving beneficial feedback from other graduate nurse participants about their performance during the simulation, including how to demonstrate empathy, support family members, and cope with the situation when an urgent problem occurred. The new graduate nurses felt that the discussion in the debriefing session provided effective learning opportunities. Overall, the participants highly praised the program and hoped to continue to interact with and learn from professionals from various disciplines in the future.

## 4. Discussion

The current results demonstrated that the interprofessional simulation program significantly enhanced the capability of new graduate nurses to integrate healthcare skills, raise humanity concerns, and deal with stressful situations. The participants expressed that the interprofessional simulation scenarios had a positive influence on their interprofessional training. New graduate nurses’ collaboration within the interprofessional team is one of the important components in their core competencies [12,32]. Currently, the training of new graduate nurses in Taiwan consists primarily of interprofessional meetings that stay within the framework of past learning experiences. The results of the present study provide evidence of positive learning outcomes through the use of an interprofessional simulation program for training new graduate nurses. A major contribution of this study is an appreciation that the interprofessional simulation program should be taken into consideration for new-graduate training. The positive learning outcomes help new graduate nurses understand the required clinical healthcare policies and traditions for working with doctors, respiratory therapists, and other health care providers when managing patients with urgent care needs [3,5,6]. This study indicated that an interprofessional simulation training program can improve the clinical care ability of new graduate nurses; increase their understanding of the roles of other healthcare professionals; enhance the understanding of interprofessional teamwork; and help participants develop their understanding of teamwork. Interprofessional simulation training can improve the ability of new graduate nurses to succeed in clinical tasks as well as increase their care capabilities and improve future retention rates.

Nursing core competence is the professional ability of nurses to attain relevant, positive outcomes in patient care. Critical thinking, assessment, communication, and technical competencies are four major competencies of new graduate nurses as established by the American Association of Colleges of Nursing. Evaluation of the nursing competencies needs to include the essential knowledge and skills related to patient care [33]. In the present study, a researcher team used the NCI to evaluate the learning outcomes of new graduate nurses engaged in the interprofessional simulation training program. The development of the nursing competence instrument was based on the core competencies and values of nursing education in Taiwan proposed by the Taiwan Nursing Accreditation Council. The results of this study demonstrated the consistency of the learning goals for the improvement of crucial nursing competencies in new graduate nurses. Furthermore, the present research showed that simulation training enhanced nursing competencies.

The International Nursing Association for Clinical Simulation and Learning (2016) states that the use of real patient scenarios provides learners the ability to apply their knowledge and make patient-care decisions for the best practice. The present study incorporated patients’ real scenarios into the simulation training. The new-graduate nurse participants in the study provided qualitative feedback that the simulations were useful for practice as well as for situations in their daily practice. This practice helps new graduate nurses function appropriately when the patient has urgent problems that require interprofessional care. Other studies also have shown that real-patient scenarios help learning [34,35], and simulation training helps learners recognize patient safety issues [23] and become familiar with the clinical setting [36]. Patient safety and familiarity with patient care are crucial issues for new graduate nurses [1,8]. The results of this study showed that the interprofessional simulation training program helped new graduate nurses close the gap of reality shock as well as establish the ability to care for complex patients with multidisciplinary healthcare teams.

Our findings have corroborated the results of several other studies. Simulation training programs have played critical roles in realistic clinical situations, such as to increase the understanding of the responsibilities of other healthcare professionals, and to encourage interprofessional caring attitudes [37,38]; to enhance critical thinking [39]; to improve interprofessional communication skills [38]; to advance the recognition of patient safety [23,40,41,42]; and to inspire learning motivation and satisfaction [43]. In China, research on the outcomes of a standardized patient-based simulation of anaphylactic shock management for 104 new graduate nurses found significant improvement in the competencies of assessment skills and ability to place the patient in a correct position, to maintain an airway, and to administer oxygen and other therapies after the simulation training [6]. In Korea, an online simulation training program helped new graduate nurses enhance their knowledge and skills, as well as improve their confidence in patient care [30]. In the United States, nursing students who attended a critical care simulation stated that the simulation allowed them, in a safe environment, to critically think through the steps of care for critical patients before the capstone rotation, and to increase confidence, critical thinking, and reasoning [44]. The application of interprofessional simulation training in many settings around the world has established that the training can enhance core competencies.

This study has limitations. First of all, it did not have a control group that had no simulation training. It did have, though, a pre-and post-test design that permitted evaluation of learning outcomes among new graduate nurses. In addition, this study was conducted over one year with one cohort of nurses in one hospital in Taiwan; thus, caution is needed in generalizing the findings to other settings. Finally, the recruitment process and training program were delayed by national and hospital policies instituted because of the COVID-19 pandemic. This resulted in there being a limited number of participants recruited and also in being unable to recruit a control group. Despite the limitations, the study has contributed to broad experience that supports interprofessional simulation programs as beneficial in the training of new graduate nurses who are entering careers in patient care.

## 5. Conclusions

This study examined the learning outcomes of an interprofessional simulation training program for new graduate nurses in Taiwan. The program significantly enhanced the ability of the nurses to perform integrated care, express their humanity concerns, and deal with the tension of nursing care. To ensure the smooth transitioning of new graduate nurses into their new roles and duties, the interprofessional simulation training program should be implemented in both the undergraduate curriculum and new graduate training in clinical settings; this policy may help nursing students become familiar early in their careers with the professional teams for patient care. The results of this study provide an important indicator for hospitals and government when making policy and in-service training for new graduate nurses. Further research should track the clinical care performance of new graduate nurses who have participated in interprofessional simulation programs.

## Figures and Tables

**Table 1 ijerph-19-13839-t001:** Rationale of the interprofessional simulation program.

Nursing Competency	Learning Objectives	Application in the Program
Care abilities	Identification of clinical problems Management of clinical problems Application of communication skills and participation in collaborative care with a multidisciplinary team member Evaluation of the care results	Participants apply the skills of assessment, management, and outcome evaluation for a patient presenting emergency condition. Participants practice communication skills and collaboration with physicians and respiratory therapists in an emergency situation.
Caring	Demonstration of compassionate care and communication skills with the involved family member	Participants practice providing care to a patient presenting emergency condition and their family members.
Advancing skills	Identification of clinical problems Management of clinical problems Participation in collaborative care with a multidisciplinary team member Evaluation of the care results	Participants apply the skills of assessment, management, and outcome evaluation for a patient requiring resuscitation.
Dealing with tension	Identification of clinical problems Management of clinical problems Evaluation of the care results	Participants experience and deal with a tense situation while the patient presenting emergency condition.

**Table 2 ijerph-19-13839-t002:** Demographic data of participants (*N* = 93).

Variable	
Gender	
Male, n (%)	6 (6.5)
Female, n (%)	87 (93.5)
Age (years) M (SD)	22.5 (0.80)
Education	
Associate degree, n (%)	4 (4.3)
Bachelor, n (%)	89 (95.7)
Department	
Medicine-Surgery, n (%)	56 (60.2)
Intensive care unit, n (%)	21 (22.6)
Emergency department, n (%)	6 (6.5)
Other departments, n (%)	10 (10.8)
Nursing experience (months), M (SD) *	7.0 (5.42)

M, median; SD, standard deviation; * min to max: 1–23 months.

**Table 3 ijerph-19-13839-t003:** Pretest and post-test core nursing values (*N* = 93).

Variable	Pretest		Post-Test		t
	M	SD	M	SD	
Dealing with tension	25.4	6.50	29.0	5.49	6.87 ***
Integrating care abilities	61.5	14.0	13.3	13.3	10.19 ***
Leading humanity concerns	43.2	9.44	48.3	8.03	6.86 ***
Advancing career talents	69.0	45.5	9.95	52.6	10.12 ***

M, mean; SD, standard deviation. *** *p* < 0.0001.

**Table 4 ijerph-19-13839-t004:** Difference in core competencies by length of nursing experience (*N* = 93).

Variable	≤6 Months (*N* = 48)	7–12 Months (*N* = 36)	13–24 Months (*N* = 9)		
	Mean Rank	Mean Rank	Mean Rank	χ^2^	*p*
Pretest Dealing with tension	40.91	48.24	74.56	11.94	0.003
Integrating care abilities	38.89	52.86	66.83	10.91	0.004
Leading humanity concerns	42.35	49.24	62.83	4.78	0.092
Advancing career talents	40.86	49.13	71.22	9.97	0.007
Post-test Dealing with tension	42.77	46.57	71.28	8.51	0.014
Integrating care abilities	40.70	48.67	73.94	11.74	0.003
Leading humanity concerns	43.46	46.81	66.67	5.63	0.060
Advancing career talents	41.58	47.24	74.94	11.61	0.003
Dealing with tension (change)	49.70	45.28	39.50	1.33	0.514
Integrating care abilities (change)	47.22	43.13	61.33	3.29	0.193
Leading humanity concerns (change)	48.03	44.35	52.11	0.74	0.689
Advancing career talents (change)	48.80	43.75	50.39	0.88	0.643
Satisfaction	46.0	45.65	57.72	1.64	0.441

## Data Availability

The data that support the findings of this study are available on request from the corresponding author, C.Y.H.

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
