# Peer review of "The Outcomes of an Interprofessional Simulation Program for New Graduate Nurses"

_ijerph, 2022, doi:10.3390/ijerph192113839_

Round 1
Reviewer 1 Report
I have reviewed the manuscript entitled “The Outcomes of an Interprofessional Simulation Program for New Graduate Nurses”. The theme developed is very interesting and original. It allows to contribute to the training and integration programs of professionals in the health area, in addition to fostering interpersonal relationships between elements of the multidisciplinary team and improving the performances and outcomes of the care provided. I believe that the readers of the journal, especially those with an interest in clinical supervision, nursing training and health care improvement, will appreciate your work as I have.
I can only suggest small notes (details):
a) In line 96-97 on page 3, name the variables in the order they were placed in table 1;
b) The Learning Satisfaction Questionnaire developed by the research team was based on research experience? Or some literature review) (e.g.: scoping review; systematic review; Delphi panel?) this information could clarify the develop process of its construction);
I have nothing to add and I wish you all the best with its publication.
Best Regards
The reviewer.
Author Response
Dear Reviewers,
Thank you very much for the opportunity to respond to the reviewer’s comments. Authors were delighted with the feedback from the reviewers and have addressed the suggested textual changes through the additions and corrections recorded in the manuscript.
|
Reviewer#1 |
Reply |
|
I have reviewed the manuscript entitled “The Outcomes of an Interprofessional Simulation Program for New Graduate Nurses”. The theme developed is very interesting and original. It allows to contribute to the training and integration programs of professionals in the health area, in addition to fostering interpersonal relationships between elements of the multidisciplinary team and improving the performances and outcomes of the care provided. I believe that the readers of the journal, especially those with an interest in clinical supervision, nursing training and health care improvement, will appreciate your work as I have. I can only suggest small notes (details): a) In line 96-97 on page 3, name the variables in the order they were placed in table 1; |
Thank you reviewer 1
Edited in Line 127-128. |
|
b) The Learning Satisfaction Questionnaire developed by the research team was based on research experience? Or some literature review) (e.g.: scoping review; systematic review; Delphi panel?) this information could clarify the develop process of its construction); I have nothing to add and I wish you all the best with its publication. |
Revised in Line151-152
Thank you very much reviewer1 |
|
Reviewer #2 |
|
|
It is necessary to indicate whether the included respondents gave any consent to be included in the study |
Added in Line 229-230 |
|
Why the authors mention Nursing Competence, Learning Satisfaction. Simulation Program and Procedure? How were these subsections chosen and why? |
Added more information in Line146-149 and 162-164 |
|
Reviewer #3 |
|
|
This is an excellent paper. It reads well, the background, methods, methodology and results are well described. The discussion and conclusions not only describe the theoretical and practical limitations of the study well, but relate this study to appropriate studies elsewhere. The lack of a control group is very common in this type of study and is acknowledged as a limitation. The paper is well constructed - as questions arise for the reader the answers are presented (as an example Table 2 provokes the question 'does the diversity within the group play a role' and this is answered almost immediately for one parameter (length of experience) in table 3). |
Thank you reviewer 3 |
|
Reviewer #4 |
|
|
The article is in line with the topic of the journal. The paper is clearly written and follows the usual scientific structure. The title, abstract and conclusions correspond to the article content. However, the authors should supplement the methodology section (2.3.2 and 2.3.3) with examples of the questionnaires statements used. |
Added in Line 130-149, 162-164, and 240-245 |
|
Reviewer 5 |
|
|
Thank you for the opportunity to review this paper. Overall, I found the paper to be well written. I would, however, suggest some minor changes as outlined below. I also found it difficult to follow the results using the Nursing Competence Instrument – I had questions on how simulation could impact continuing professional development and compassionate care. Perhaps these need to be elaborated on in the methods section, or a detailed discussion provided later in the paper.
|
Added more information related to the tools and the rationale for using in the method section in Line130-149, 162-164, and 240-245 |
|
Page 1, line 41: The del Bueno paper is a little dated. Would also like to see setting and sample number for this study.
|
Replaced with other reference in Line 41-44. |
|
Page 2, line 72: Both sentences feel isolated. Could more detail be provided on simulation training here rather than this being a one-line statement? A link to evidence would be good as well. |
Revised this paragraph and provided more details about the simulation training in Line 79-91 |
|
Page 3, line 116: Likert is spelt incorrectly.
|
Corrected in Line155 |
|
Page 3, line 124: The opening line here seems redundant. You could start the paragraph with “Clinical cases...”
|
Corrected in Line166 |
|
Page 5, line 191: Table two – the meaning of ‘advancing career talents’ is not clear to me. Although mentioned earlier in the section on the instruments used, I think you need to expand so this is clear to the reader. I am also unsure how this relates to the use of simulation. |
Added more information related to the tools and the rationale for using in the method section in Line130-149, 162-164, and 240-245 |
|
Reviewer 6 |
|
|
The manuscript entitled The Outcomes of an Interprofessional Simulation Program for New Graduate Nurses explores the learning outcomes of an interprofessional simulation program in new graduate nurses during their training program. The topic is meaningful and interesting, and it meets the scope of the journal. However, the research design and the research methods are bad. Any kind of training can improve the participants’ abilities. Without comparison, we cannot know which kind of training method works well. The nursing core competencies cannot be valued only by a self-report investigation. On current situation, the manuscript cannot be published. It must be re-designed. |
Thank you very much for the comments. This study was forced to finalized due to the COVID-19 pandemic that resulted in no control group. However, this study has contributed experiences that supports the benefits of interprofessional simulation program for new graduate nurses. It has mentioned in the limitation section in Line 376-381. |

Reviewer 2 Report
it is necessary to indicate whether the included respondents gave any consent to be included in the study
Why the authors mention Nursing Competence, Learning Satisfaction. Simulation Program and Procedure? How were these subsections chosen and why?
Author Response
Dear Reviewers
Thank you very much for the opportunity to respond to the reviewer’s comments. Authors were delighted with the feedback from the reviewers and have addressed the suggested textual changes through the additions and corrections recorded in the manuscript.
|
Reviewer#1 |
Reply |
|
I have reviewed the manuscript entitled “The Outcomes of an Interprofessional Simulation Program for New Graduate Nurses”. The theme developed is very interesting and original. It allows to contribute to the training and integration programs of professionals in the health area, in addition to fostering interpersonal relationships between elements of the multidisciplinary team and improving the performances and outcomes of the care provided. I believe that the readers of the journal, especially those with an interest in clinical supervision, nursing training and health care improvement, will appreciate your work as I have. I can only suggest small notes (details): a) In line 96-97 on page 3, name the variables in the order they were placed in table 1; |
Thank you reviewer 1
Edited in Line 127-128. |
|
b) The Learning Satisfaction Questionnaire developed by the research team was based on research experience? Or some literature review) (e.g.: scoping review; systematic review; Delphi panel?) this information could clarify the develop process of its construction); I have nothing to add and I wish you all the best with its publication. |
Revised in Line151-152
Thank you very much reviewer1 |
|
Reviewer #2 |
|
|
It is necessary to indicate whether the included respondents gave any consent to be included in the study |
Added in Line 229-230 |
|
Why the authors mention Nursing Competence, Learning Satisfaction. Simulation Program and Procedure? How were these subsections chosen and why? |
Added more information in Line146-149 and 162-164 |
|
Reviewer #3 |
|
|
This is an excellent paper. It reads well, the background, methods, methodology and results are well described. The discussion and conclusions not only describe the theoretical and practical limitations of the study well, but relate this study to appropriate studies elsewhere. The lack of a control group is very common in this type of study and is acknowledged as a limitation. The paper is well constructed - as questions arise for the reader the answers are presented (as an example Table 2 provokes the question 'does the diversity within the group play a role' and this is answered almost immediately for one parameter (length of experience) in table 3). |
Thank you reviewer 3 |
|
Reviewer #4 |
|
|
The article is in line with the topic of the journal. The paper is clearly written and follows the usual scientific structure. The title, abstract and conclusions correspond to the article content. However, the authors should supplement the methodology section (2.3.2 and 2.3.3) with examples of the questionnaires statements used. |
Added in Line 130-149, 162-164, and 240-245 |
|
Reviewer 5 |
|
|
Thank you for the opportunity to review this paper. Overall, I found the paper to be well written. I would, however, suggest some minor changes as outlined below. I also found it difficult to follow the results using the Nursing Competence Instrument – I had questions on how simulation could impact continuing professional development and compassionate care. Perhaps these need to be elaborated on in the methods section, or a detailed discussion provided later in the paper.
|
Added more information related to the tools and the rationale for using in the method section in Line130-149, 162-164, and 240-245 |
|
Page 1, line 41: The del Bueno paper is a little dated. Would also like to see setting and sample number for this study.
|
Replaced with other reference in Line 41-44. |
|
Page 2, line 72: Both sentences feel isolated. Could more detail be provided on simulation training here rather than this being a one-line statement? A link to evidence would be good as well. |
Revised this paragraph and provided more details about the simulation training in Line 79-91 |
|
Page 3, line 116: Likert is spelt incorrectly.
|
Corrected in Line155 |
|
Page 3, line 124: The opening line here seems redundant. You could start the paragraph with “Clinical cases...”
|
Corrected in Line166 |
|
Page 5, line 191: Table two – the meaning of ‘advancing career talents’ is not clear to me. Although mentioned earlier in the section on the instruments used, I think you need to expand so this is clear to the reader. I am also unsure how this relates to the use of simulation. |
Added more information related to the tools and the rationale for using in the method section in Line130-149, 162-164, and 240-245 |
|
Reviewer 6 |
|
|
The manuscript entitled The Outcomes of an Interprofessional Simulation Program for New Graduate Nurses explores the learning outcomes of an interprofessional simulation program in new graduate nurses during their training program. The topic is meaningful and interesting, and it meets the scope of the journal. However, the research design and the research methods are bad. Any kind of training can improve the participants’ abilities. Without comparison, we cannot know which kind of training method works well. The nursing core competencies cannot be valued only by a self-report investigation. On current situation, the manuscript cannot be published. It must be re-designed. |
Thank you very much for the comments. This study was forced to finalized due to the COVID-19 pandemic that resulted in no control group. However, this study has contributed experiences that supports the benefits of interprofessional simulation program for new graduate nurses. It has mentioned in the limitation section in Line 376-381. |

Reviewer 3 Report
This is an excellent paper. It reads well, the background, methods, methodology and results are well described. The discussion and conclusions not only describe the theoretical and practical limitations of the study well, but relate this study to appropriate studies elsewhere. The lack of a control group is very common in this type of study and is acknowledged as a limitation. The paper is well constructed - as questions arise for the reader the answers are presented (as an example Table 2 provokes the question 'does the diversity within the group play a role' and this is answered almost immediately for one parameter (length of experience) in table 3).
Author Response

(The authors gave the same response as above.)

Reviewer 4 Report
The article is in line with the topic of the journal. The paper is clearly written and follows the usual scientific structure. The title, abstract and conclusions correspond to the article content. However, the authors should supplement the methodology section (2.3.2 and 2.3.3) with examples of the questionnaires statements used.
Author Response

(The authors gave the same response as above.)

Reviewer 5 Report
Thank you for the opportunity to review this paper. Overall, I found the paper to be well written. I would, however, suggest some minor changes as outlined below. I also found it difficult to follow the results using the Nursing Competence Instrument – I had questions on how simulation could impact continuing professional development and compassionate care. Perhaps these need to be elaborated on in the methods section, or a detailed discussion provided later in the paper.
Page 1, line 41: The del Bueno paper is a little dated. Would also like to see setting and sample number for this study.
Page 2, line 72: Both sentences feel isolated. Could more detail be provided on simulation training here rather than this being a one-line statement? A link to evidence would be good as well.
Page 3, line 116: Likert is spelt incorrectly.
Page 3, line 124: The opening line here seems redundant. You could start the paragraph with “Clinical cases...”
Page 5, line 191: Table two – the meaning of ‘advancing career talents’ is not clear to me. Although mentioned earlier in the section on the instruments used, I think you need to expand so this is clear to the reader. I am also unsure how this relates to the use of simulation.
Author Response

(The authors gave the same response as above.)

Reviewer 6 Report
The manuscript entitled The Outcomes of an Interprofessional Simulation Program for New Graduate Nurses explores the learning outcomes of an interprofessional simulation program in new graduate nurses during their training program. The topic is meaningful and interesting, and it meets the scope of the journal. However, the research design and the research methods are bad. Any kind of training can improve the participants’ abilities. Without comparison, we cannot know which kind of training method works well. The nursing core competencies cannot be valued only by a self-report investigation. On current situation, the manuscript cannot be published. It must be re-designed.
Author Response

(The authors gave the same response as above.)

Round 2
Reviewer 6 Report
As said in last review, the research design and the research methods are bad. A metion of limitation cannot sovle the pronlem of the research design. Frankly speaking, this is not a scientific study. Any kind of training can improve the participants’ abilities. Without comparison, we cannot know which kind of training method works well. The nursing core competencies cannot be valued only by a self-report investigation. Although the author On current situation, the manuscript cannot be published. It must be re-designed.
Author Response
Response to Reviewer Comments
Thank you very much for the opportunity to respond to the reviewer’s comments. Authors were delighted with the feedback from the reviewers and have addressed the suggested textual changes through the additions and corrections recorded in the manuscript.
|
Reviewer 6 |
Reply |
|
As said in last review, the research design and the research methods are bad. A metion of limitation cannot sovle the pronlem of the research design. Frankly speaking, this is not a scientific study. Any kind of training can improve the participants’ abilities. Without comparison, we cannot know which kind of training method works well. The nursing core competencies cannot be valued only by a self-report investigation. Although the author On current situation, the manuscript cannot be published. It must be re-designed. |
Thank you very much for the Reviewer comments. Authors have revised the research method section. The revision added more details on the rational and application of interprofessional simulation program and measurement in Line120-126, Line 141-187and Line 216-222. The revised results present the outcomes of interprofessional simulation training program for new graduate nurses from quantitative and qualitative perspectives in Line356-450 |
